# Role of the Nephrologist in Non-Kidney Solid Organ Transplant (NKSOT)

**DOI:** 10.3390/healthcare11121760

**Published:** 2023-06-15

**Authors:** Iris Viejo-Boyano, Luis Carlos López-Romero, Luis D’Marco, Ana Checa-Ros, María Peris-Fernández, Enrique Garrigós-Almerich, María Carmen Ramos-Tomás, Ana Peris-Domingo, Julio Hernández-Jaras

**Affiliations:** 1Nephrology Department, Hospital Universitari i Politècnic La Fe, 46026 Valencia, Spain; 2Nephrology Department, Hospital General Universitario de Valencia, 46014 Valencia, Spain; 3Grupo de Investigación en Enfermedades Cardiorrenales y Metabólicas, Departamento de Medicina y Cirugía, Facultad de Ciencias de la Salud, Universidad Cardenal Herrera-CEU, 46115 Valencia, Spain

**Keywords:** chronic kidney disease, end-stage renal disease, solid organ transplant, nephrology

## Abstract

Background: Chronic kidney disease (CKD) is a common complication of a non-kidney solid organ transplant (NKSOT). Identifying predisposing factors is crucial for an early approach and correct referral to nephrology. Methods: This is a single-center retrospective observational study of a cohort of CKD patients under follow-up in the Nephrology Department between 2010 to 2020. Statistical analysis was performed between all the risk factors and four dependent variables: end-stage renal disease (ESKD); increased serum creatinine ≥50%; renal replacement therapy (RRT); and death in the pre-transplant, peri-transplant, and post-transplant periods. Results: 74 patients were studied (7 heart transplants, 34 liver transplants, and 33 lung transplants). Patients who were not followed-up by a nephrologist in the pre-transplant (*p* < 0.027) or peri-transplant (*p* < 0.046) periods and those who had the longest time until an outpatient clinic follow-up (HR 1.032) were associated with a higher risk of creatinine increase ≥50%. Receiving a lung transplant conferred a higher risk than a liver or heart transplant for developing a creatinine increase ≥50% and ESKD. Peri-transplant mechanical ventilation, peri-transplant and post-transplant anticalcineurin overdose, nephrotoxicity, and the number of hospital admissions were significantly associated with a creatinine increase ≥50% and developing ESKD. Conclusions: Early and close follow-up by a nephrologist was associated with a decrease in the worsening of renal function.

## 1. Introduction

Chronic kidney disease (CKD) is a common complication of a non-kidney solid organ transplant (NKSOT), causing increased morbidity and mortality in this group of patients. The increase in overall survival due to advancements in surgical techniques and immunosuppression management has promoted growth in the absolute number of patients who develop these complications [1,2].

A large study published in 2003 by Ojo AO et al. [3], with a population cohort of 69,321 patients who received NKSOT between 1990 and 2000, presented exciting results on the predisposing factors for CKD of each type of NKSOT. In this study, 11,426 patients (16.5%) developed end-stage kidney disease (ESKD) during a median follow-up of 36 months. The cumulative incidence range varied from 6.9% for those with heart-lung transplants to 21.3% for those with intestine transplants. Currently, the absolute number of transplants performed has increased worldwide [4,5,6]. Together with improved surgical techniques and more effective immunosuppressive regimens [7,8,9], this means that new revisions on the complications and risk factors, specifically on ESKD, should be carried out.

Generally, there are pre-transplant, peri-transplant, and post-transplant predisposing factors in the middle- and long-term periods. However, specific characteristics in each type of transplant give different risks to developing CKD. In a series of renal biopsies from patients with NKSOT, variations in histological pattern were observed, demonstrating differences between the types of organ transplantation. For example, arteriolar hyalinosis was more common in heart and lung transplant recipients than in liver transplant recipients, while primary glomerular disease was more common in the latter [10].

The knowledge of all the general and specific predisposing factors of each type of transplant may allow an early approach and minimize the progressive worsening of renal function and its short- and long-term complications. CKD is currently the eighth leading cause of death in developed countries, such as Spain. However, its incidence continues to grow, and it is estimated that it will be the second leading cause of death in Spain by 2100, after Alzheimer’s disease [11]. This incidence increases when there are risk factors to develop it, as is the case for patients with NKSOT. Thus, to identify these risk factors and address them at an early stage is crucial; we believe that the nephrologist plays an important role in these interventions. Therefore, this study aimed to analyze the nephrologist’s contribution to managing patients with NKSOT.

## 2. Materials and Methods

This is a single-center retrospective observational study on a cohort of CKD patients under follow-up in the Department of Nephrology at La Fe University and Politècnic Hospital in Valencia (Spain) from January 2010 to December 2020.

This study was performed in accordance with the ethical principles of the Declaration of Helsinki as revised in 2013. The study belonged to the project “Enfermedad renal crónica en trasplante de órgano sólido no renal. Análisis de la incidencia, factores de riesgo y pronóstico global de los diferentes tipos de trasplante de órgano sólido no renal”, approved by the Hospital Universitari i Politècnic La Fe Clinical Research Ethics Committee, with the reference number 2021-302-1.

### 2.1. Study Population

The cohort of patients was extracted from our hospital’s nephrology outpatient clinic and the End Stage Renal Disease Unit records, with a total of 212 patients: 44 heart transplant recipients, 93 liver transplant recipients, and 75 lung transplant recipients. Patients transplanted before 2010 (101 patients) or with combined renal transplantation (9 patients), and those with no follow-up in the outpatient clinic (28 patients) were excluded. A final population of 74 patients, 7 with a heart transplant, 34 with liver transplants, and 33 with a lung transplant, was obtained (Figure 1). Subjects older than 18 years at the onset of the follow-up in the outpatient clinic or End Stage Renal Disease Unit, NKSOT recipients between 2010 and 2020, and those with a CKD-EPI estimated glomerular filtration rate (eGFR) <60 mL/min/1.73 m^2^ at baseline were included.

Patients who met the inclusion/exclusion criteria were evaluated retrospectively in three different periods (Figure 2):Pre-transplant period: Clinical and analytical variables present before the solid organ transplant until day 0 of the transplant.Peri-transplant period: From the transplant to hospital discharge.Post-transplant period: From hospital discharge to one year of follow-up in nephrology consultations.

### 2.2. Data Collection

Baseline demographics, including age, gender, major comorbidities, and etiology of kidney disease, etiology of disease associated with the transplantation, and clinical and analytical data in the different study periods (Figure 2) were collected from the medical records. Laboratory parameters included creatinine, eGFR, proteinuria, glucose, HbA1c, total cholesterol, low-density lipoprotein (LDL) cholesterol, high-density lipoprotein (HDL) cholesterol, and uric acid were determined in the hospital biochemistry laboratory by assays following Good Laboratory Practice standards. The KDIGO guidelines were adopted to manage CKD from the start of follow-up in nephrology.

### 2.3. Statistical Analysis

Statistical analysis was performed to assess the associations between all the risk factors and the four dependent variables: ESKD, increased serum creatinine ≥50%, renal replacement therapy, and death. For categorical risk factors, such as the type of transplant, we employed the Fisher’s exact test when the cell frequencies in the contingency table were less than 5, and the chi-square test for larger sample sizes. Continuous risk factors were assessed for normality using the Kolmogorov–Smirnov test. If the continuous variables followed a normal distribution, we utilized the Student’s *t*-test. Alternatively, if normality assumptions were violated, we employed the Mann–Whiney U test. To evaluate the increased risk of developing one of the categorical dependent variables, we conducted multinomial regression analyses.

Furthermore, we examined associations between normally distributed continuous variables and the dependent variables using the Pearson correlation coefficient. For continuous variables that did not follow a normal distribution, we used the Spearman correlation coefficient. Additionally, we utilized the Pearson correlation coefficient as a measure of association (also known as the coefficient of contingency) for categorical data.

The Kaplan–Meier statistic was used for survival analysis. In the Kaplan–Meier analysis, the outcome variables examined were the development of advanced ESKD and the need for renal replacement therapy, with the independent variable being the type of transplant. Censoring criteria were applied to individuals who did not develop ESKD or renal replacement therapy and completed the follow-up period. The mean follow-up duration was 59.61 months, with a maximum follow-up time of 142 months.

A two-sided *p*-value < 0.05 was considered statistically significant. Analyses were performed with IBM SPSS^®^ Statistics version 26.

### 2.4. Data Availability

The datasets used and/or analyzed during the current study are available from the corresponding author upon request.

## 3. Results

### 3.1. Baseline Characteristics

The baseline characteristics are summarized in Table 1. The average age at the time of transplantation was 54.39 years. Fifty-four patients were males (73%). Recipients of heart, liver, and lung transplants accounted for 9.5% (7 patients), 45.9% (34 patients), and 44.6% (33 patients), respectively.

In the pre-transplant period, 37.8% of patients were reported to have hypertension, 27.1% had diabetes mellitus, 25.7% had dyslipidemia, 14.9% had hyperuricemia, 23% were overweight, 18.9% were obese, 55.4% were ex-smokers, 8.1% were smokers, 6.8% had cardiorenal syndrome, and 18.9% suffered from hepatorenal syndrome. The pre-transplantation eGFR was <60 mL per minute per 1.73 m^2^ (CKD) in 23% of the patients, and only 14.9% of the sample were followed-up by the nephrologist. The differences between transplants are summarized in Table 1.

In the peri-transplant period, 58.1% of the patients presented acute kidney injury (AKI), 5.4% required RRT, and only 18.9% were under nephrology follow-up. During their hospital admission, 51.4% of the patients showed surgical complications and 87.8% required blood transfusions. A total of 81.8% of the patients received tacrolimus (a subset of them initially started with cyclosporine but later switched to tacrolimus during the course of their treatment); 98.6% of the sample received mycophenolate, and 90.5% of the patients received corticosteroids. Changes into the initial immunosuppression were made in nearly 15% of the patients, fundamentally motivated by renal failure. A total of 20.3% of the patients presented graft rejection, 50% overdosed on calcineurin inhibitors for their reference levels, and 64.9% received nephrotoxic agents (NSAIDs, antimicrobials, intravenous contrast, or a combination of them).

In the post-transplant period, 23% of the patients developed hypertension, 23% had post-transplant diabetes mellitus (PTDM), and 17.5% had dyslipidemia. The mean number of hospital admissions was near 7, ranging from a minimum of 1 to a maximum of 23. A total of 78.4% developed infections; 66.2% of the patients presented an overdose of calcineurin inhibitors in more than three determinations; 68.9% required changes in immunosuppression due to adverse effects, including worsening of renal function; and 31.1% presented graft rejection. In addition, 77% of the sample received nephrotoxic drugs, mostly in combination. Finally, near 10% of the sample presented secondary thrombotic microangiopathy (TMA). The differences between transplants are summarized in Table 1.

The mean time from the transplant to the start of follow-up in the nephrology outpatient clinic was 33 months. The mean eGFR was 39.24 mL/min/1.73 m^2^. From the start of outpatient follow-up, renal function worsened in 45.9% of the sample, versus 54.1% who improved or maintained stability. Among the total sample, 45 patients (59.5%) experienced a 50% increase in their baseline creatinine levels, 24 patients (32.43%) reached ACKD (GFR < 30 mL/min/1.73 m^2^), 8 patients (10.8%) ended up in RRT, and 21 patients (28.4%) died. Table 1 provides a comprehensive summary of the differences observed among the various transplant types.

### 3.2. Increased Baseline Creatinine by 50%

Baseline creatinine levels increased by 50% in 45 patients. The mean age of these patients was 36.55 years, and most of them (75%) were males. The type of transplant was significantly associated with a 50% increase in baseline creatinine (*p* < 0.05). The highest risk was observed among lung transplant recipients, followed by heart transplant recipients, and finally, liver transplant recipients.

Significant associations were observed between a 50% increase in baseline serum creatinine levels and various factors, such as the absence of previous CKD, peri-transplant mechanical ventilation, peri-transplant calcineurin inhibitor overdose, peri-transplant total nephrotoxic exposure, peri-transplant antimicrobials nephrotoxicity, peri-transplant intravenous contrast administration, absence of everolimus usage, no changes in immunosuppression, post-transplant calcineurin inhibitor overdose, post-transplant intravenous contrast administration, and the number of hospital admissions.

Regarding nephrotoxic factors during the peri-transplant period, after performing a multinomial regression analysis adjusted for other factors, the use of nephrotoxic antimicrobials did not retain its statistical significance, while the association with intravenous contrast administration remained significant, indicating a 3.16-fold increased risk of a 50% rise in serum creatinine levels. Detailed results of statistically significant associations are summarized in Table 2. Among the variables analyzed within each transplant group, statistically significant associations (*p* < 0.05) were observed for lung transplant recipients with the absence of everolimus usage and the absence of changes in immunosuppression during the peri-transplant period, as well as hypertension in the post-transplant period.

Not being followed-up by nephrology in the pre- and peri-transplant period were both significantly associated with the increased baseline serum creatinine by 50%. A delay in the follow-up by a nephrologist (each month), was associated with a 1.03-fold increased risk of increased serum creatinine.

### 3.3. End-Stage Kidney Disease (ESKD)

Twenty-four patients reached ESKD. The mean age was 55.63 years, and most of them (66.7%) were men. The type of organ transplanted was significantly associated with reaching ESKD (*p* < 0.05). Lung transplant recipients were at the highest risk. Compared with them, receiving a liver transplant presented a 6.94 times lower risk (HR 0.14 [95% CI] 0.045 to 0.463) and receiving a heart transplant, 7.19 times lower, although the latter was not significantly lower (HR 0.13 [95% CI] 0.015 to 1.28).

Peri-transplant mechanical ventilation (*p* < 0.035), peri-transplant (*p* < 0.015) and post-transplant (*p* < 0.037) anticalcineurin overdose, peri-transplant (*p* < 0.028) and post-transplant nephrotoxic antimicrobials (*p* < 0.032), peri-transplant intravenous contrast (*p* < 0.004), and the number of hospital admissions (*p* < 0.02) were significantly associated with the worsening of renal function. In relation to nephrotoxic associated factors in the peri-transplant period, after doing a multinomial regression adjusted for the rest of the factors, the use of nephrotoxic antimicrobials lost its statistical significance; however, intravenous contrast showed an increased risk (3.62 times) to develop ESKD. Statistically significant results are summarized in Table 3.

Regarding those lung transplant recipients, dyslipidemia in the post-transplant period was significantly associated with the development of ESKD (*p* < 0.05).

Mean ESKD-free survival was 93.29 months (95% CI 79.04–107.5), 121.5 months in heart transplantation recipients (95% CI of 86.70–155.297), 104 months in liver transplantation recipients (95% CI of 86.67–122.134), and 66.86 months in lung transplantation recipients (CI at 95% of 55.93–80.79) (Figure 3).

### 3.4. Renal Replacement Therapy (RRT)

From the study population, 8 patients required RRT. The mean age of these patients was 48 years, and most of them (75%) were men. It was significantly associated with the overdose of calcineurin inhibitors in the peri-transplant period (*p* < 0.046), with an 8.4-fold increased risk (HR 8.4 [95% CI] 1.01 to 72.155). The median survival without reaching RRT was 124.93 months [95% CI] 112.42–137.43. This differs between transplants: 142 months in heart transplant, 118 months in liver transplant [95% CI] 107.94–129.67, and 104 months in lung transplant [95% CI] 92.85–115.88 (Figure 3). However, these differences were not statistically significant when comparing all the groups.

### 3.5. Death

Twenty-one of the patients died during the follow-up period. The mean age at transplant was 54.25 years, most of them (76.2%) were men. The type of organ transplanted was not statistically associated with any of the exitus variables. It was significantly associated with active smoking in the pre-transplant (*p* < 0.028) period and the number of hospital admissions in the post-transplant (*p* < 0.006) period, with a 1.18-fold increased risk of death for each hospital admission (HR 1.118 [95% CI] 1.026 to 1.217).

## 4. Discussion

NKSOT has expanded in number and complexity in recent years. The increase in knowledge and experience allows medical transplant teams to improve patient care, decreasing comorbidities and mortality. Many of these comorbidities affect renal function, generating a progressive growth of patients who develop CKD among NKSOT recipients and placing nephrology care as a critical component in managing these patients.

The strategies to minimize CKD must begin at the pre-transplant stage; treating cardiovascular and renal risk factors; followed by peri-transplantation monitoring and management, treating AKI, avoiding nephrotoxicity, and adjusting immunosuppression; and finally, in the post-transplantation stage, treating CKD progression factors, as well as nephroprotection and the adjustment of immunosuppressants [11].

Knowing which factors are closely related to increasing susceptibility and progression of CKD and its complications allows for identifying patients at higher risk, and an early referral to the nephrology department may help.

The risk of developing CKD and complications differs among the type of transplant performed. In a study from Ojo AO et al. [3], the incidence of ESKD at five years ranged from 10.9% in heart transplants to 15.8% in lung transplants and 18.1% in liver transplants. In our study, lung transplants had the highest risk of increasing their baseline creatinine by 50% and developing ESKD. It is noteworthy that despite heart transplant recipients being the next at risk of increasing their creatinine by 50%, this does not occur for the development of ESKD, in which liver transplants present a higher risk than those with heart transplants. These findings could be related to the baseline of renal function in liver transplant patients. Thus, the renal function was worse, with 26.6% of patients with stage G3b and 13.4% with stage G4; in the case of heart transplant recipients, all were in a G3a stage. The need for RRT also occurs earlier in lung transplant recipients, followed by liver transplant recipients, and finally those with a heart transplant.

One of the remarkable findings in our study was the low rate of follow-up by the nephrology department despite CKD. In the literature, joint follow-up with a nephrologist has been associated with fewer cardiovascular events in liver transplant patients [12] and better management both before and after liver [13], heart, and lung [14] transplantation.

In our study, we found that for those patients in the pre-transplant period, 23% had documented CKD, but nephrology had previously followed up only in 14.9%. This proportion varies and depends on the type of transplant. Therefore, all heart transplant recipients with CKD before transplantation were followed for nephrologists, against less than half of liver transplant recipients. In the cases of lung transplants, no patient had previous CKD, but two patients underwent nephrological follow-up for episodes of recovered AKI. Not having a prior nephrology follow-up was associated with a 4.97 times higher risk of increasing baseline creatinine by 50%.

During the peri-transplant period, 58.1% of the patients presented AKI, but just 18.9% of them had a nephrology follow-up. Once again, this proportion changes depending on the type of transplant, with nephrology monitoring half of the heart transplant recipients who had AKI, one-third of the liver transplants, and a quarter of those with lung transplants. Not having a peri-transplant follow-up was associated with a 3.34-fold increased risk of increasing baseline creatinine by 50%.

The mean time from the transplantation to the beginning of the nephrology outpatient evaluation and follow-up was 33.01 months, with a shorter time for those with liver transplant (28.24 months), followed by heart transplant recipients (33.71 months), and ending with lung transplant recipients (38.64 months). The delay in follow-up was significantly associated with a 50% increase in baseline creatinine, with an increased risk of developing a renal event of 1.03 times for each month without a follow-up.

In the literature, several studies have shown that pre-transplant kidney damage increased the risk of CKD [3,15,16]. In heart transplant recipients, kidney damage is related to cardiovascular risk factors and renal hypoperfusion in the form of cardiorenal syndrome. In liver transplant recipients, it is also associated with renal hypoperfusion in the form of hepatorenal syndrome, acute tubular necrosis, and glomerulonephritis. Regarding lung transplant recipients, previous renal failure is less frequent, and its etiology is the same as for the general population.

In our study, we observed no significant cardiovascular risk factor associated with the development of ESKD. Nevertheless, active smoking was associated with mortality.

Similarly, investigations have reported that peri-transplant kidney damage is significantly related to developing CKD [3,17,18]. Thus, risk factors were similar in heart and liver transplantation, attributed to surgical complications and hypovolemia secondary to surgery (need for vasoactive drugs and transfusions), as well as the use of nephrotoxic medications and overdose of calcineurin inhibitors [17,18]. For those patients with lung transplantation, the use of nephrotoxic antimicrobial agents such as aminoglycosides and amphotericin, the need for high-dose calcineurin inhibitors due to an increased risk of acute graft rejection, and the use of mechanical ventilation are emphasized [19,20].

Analyzing our results from this period, the requirement for mechanical ventilation was associated with the development of ESKD (*p* < 0.035) and a 50% increase in serum creatinine (*p* < 0.001), with an increased risk of 2.95 and 6.36 times, respectively.

The overdose of calcineurin inhibitors was associated with the development of ESKD, 50% increase in baseline creatinine, and RRT needed, with an increased risk by 3.63, 4.08, and 8.4 times, respectively. Not using everolimus as the immunosuppressive treatment and/or adjusting the changes in immunosuppression drugs were also associated with a 50% increase in baseline creatinine. This is because switching immunosuppressants is usually associated with the development of AKI. When examining these variables within each transplant group, statistical significance was maintained only for the absence of everolimus usage and the absence of changes in immunosuppression among lung transplant recipients.

Finally, nephrotoxic antimicrobials and intravenous contrast were associated with the risk of developing ESKD and an increase of 50% in baseline creatinine. However, in the multinomial regression adjusted for the rest of the factors, this variable lost statistical significance; on the contrary, for intravenous contrast, an increased risk of 3.62 and 3.16 times for the development of ESDR and increased 50% of creatinine was observed.

Regarding the kidney damage in the post-transplant period, study findings mainly attributed to chronic damage from calcineurin inhibitors for all transplants. For heart transplants, in addition to calcineurin inhibitors, the development of cardiovascular risk factors and nephrotoxicity also contributed to the development of ESKD [14]. To reduce calcineurin inhibitors, nephrotoxicity, dose reduction protocols, and changes in immunosuppression are being studied with satisfactory results. Low-dose everolimus associated with low-dose calcineurin inhibitors in combination with mycophenolate and steroids has been associated with improved renal function, less vascular disease, and similar cardiac function at one year of follow-up studies [14,21,22]. In the case of liver transplants, it is estimated that in approximately 50% of patients who develop CKD, calcineurin inhibitor nephrotoxicity is the leading cause. Adding this medication contributes to metabolic syndrome [18,23]. Anticalcineurin drug toxicity is the leading cause of renal dysfunction for lung transplants due to the high need for immunosuppression [14,20,23].

In the post-transplant period, no cardiovascular risk factor was associated with the studied variables. However, analyzing each type of transplant, for lung transplant recipients, the dyslipidemia variable was associated with the development of ESKD; moreover, hypertension was associated with a 50% increase in baseline creatinine.

The number of admissions was significantly associated with the development of ESKD, a 50% increase in creatinine, and death, with an increased risk for each admission of 1.103, 1.16, and 1.118, respectively.

Regarding immunosuppression, an overdose of calcineurin inhibitors on more than three occasions was associated with developing ESKD and with an increase of 50% of baseline creatinine, with an augmented risk of 3.62 and 4.44 times, respectively.

Nephrotoxic drug use (nephrotoxic antimicrobials) was significantly associated with progression to ESKD. Nonetheless, this variable lost significance when adjusted for other variables of risk, although it maintains a 2.6-fold increased risk of developing it. Similarly, it occurs with the use of intravenous contrast for a 50% increase in baseline creatinine. When adjusting for other nephrotoxic drugs, despite having a 2.59-fold increased risk, it loses significance.

Two main limitations should be considered when interpreting our findings. Firstly, the small sample size of 74 patients in this study may restrict the generalizability of the results. Larger sample sizes are crucial to enhance statistical power and improve the representativeness of the findings. Secondly, the single-center design of the study introduces potential biases and may limit the external validity of the results. Replicating the study in multiple centers will strengthen the overall findings and increase their generalizability to a broader population.

## 5. Conclusions

Currently, there is a growing increase in the number and comorbidity of patients who receive a NKSOT; the multidisciplinary management of these patients is of vital importance, especially with the collaboration of different hospital departments and units. Thus, an early follow-up by a nephrologist is associated with a decrease in the decline of renal function, mainly, by acting on the renal-specific risk factors, such as the nephrotoxicity related to calcineurin inhibitor overdose, preventing AKI, and allowing the identification of patients at higher risk during the pre-surgical period. Of interest, those patients requiring mechanical ventilation during peri-transplantation or those with the highest number of hospital admissions need special attention and monitoring by the nephrology department. Further studies are required to evaluate these findings.

## Figures and Tables

**Figure 1 healthcare-11-01760-f001:**
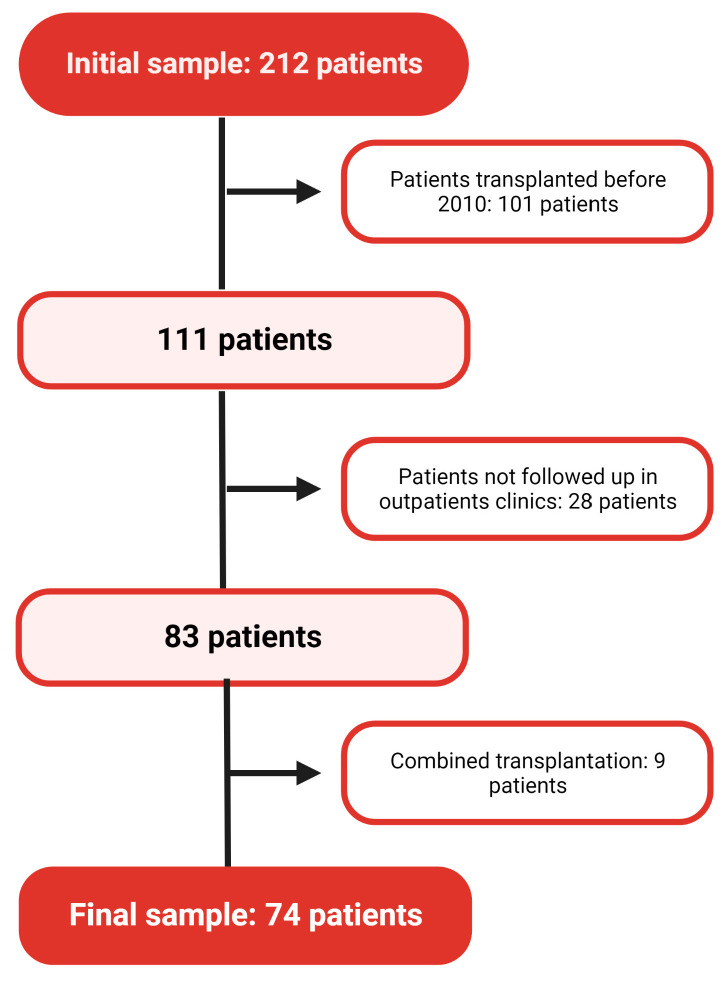
Flowchart of patient selection.

**Figure 2 healthcare-11-01760-f002:**
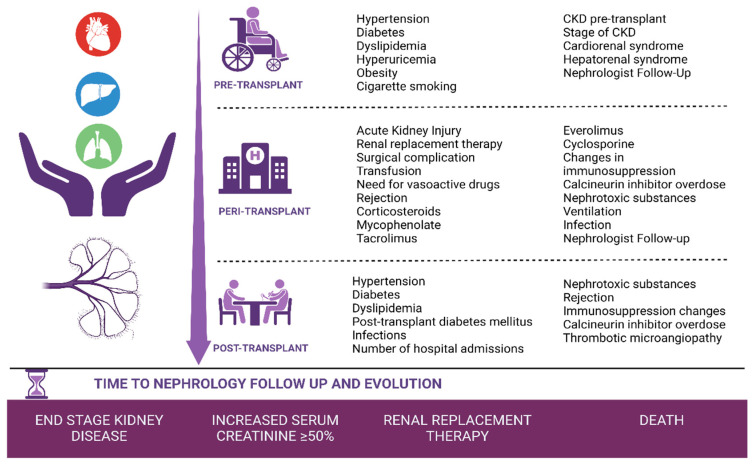
Variables analyzed in each period.

**Figure 3 healthcare-11-01760-f003:**
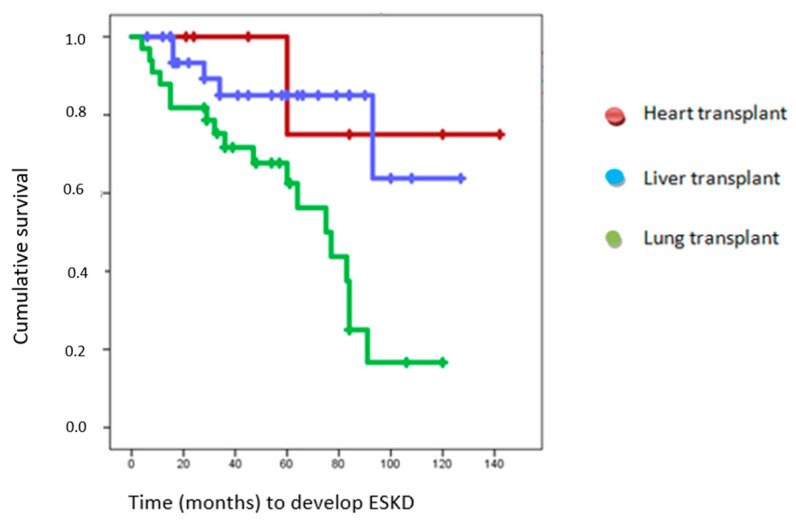
Mean ESKD-free survival of heart transplant recipients, liver transplant recipients, and lung transplant recipients. Kaplan–Meier survival analysis demonstrated a mean survival time without ESKD of 121.5 months (95% CI: 86.70–155.297) for heart transplant recipients, 104 months (95% CI: 86.67–122.134) for liver transplant recipients, and 66.86 months (95% CI: 55.93–80.79) with a median survival time of 75 months (95% CI: 52.83–97.16) for lung transplant recipients. These results showed a statistically significant association with *p* = 0.004.

**Table 1 healthcare-11-01760-t001:** Baseline characteristics. Percentages of each event (%) and mean for continuous variables.

	Global (74)	Heart (7)	Liver (34)	Lung (33)
Age (years)	54.39	49.42	58.71	51
Sex (%)	Males: 73	Males: 85.7	Males: 79.4	Males: 63.6
	Females: 27	Females: 14.3	Females: 20.6	Females: 36.4
**Pre-transplantation period**				
Hypertension (%)	37.8	57.1	39.4	33.3
Diabetes (%)	27.1	14.3	51.5	26.1
Dyslipidemia (%)	25.7	42.9	18.2	27.3
Hyperuricemia (%)	14.9	14.3	15.2	15.2
Overweight (%)	23	28.6	33.3	12.1
Obesity (%)	18.9	28.6	27.3	9.1
Ex-smoker (%)	55.4	42.9	42.4	72.7
Smoker (%)	8.1	28.6	12.1	0
Cardiorenal syndrome (%)	6.8	28.6	3	6.1
Hepatorenal syndrome (%)	18.9	0	39.4	0
CKD (%)	23	28.6	45.5	0
Stage or CKD (%)	G3a: 65.21	G3a: 100	G3a: 60	G3a: 0
	G3b: 23.91	G3b: 0	G3b: 26.6	G3b: 0
	G4: 10.88	G4: 0	G4: 13.4	G4: 0
Nephrologist Follow-up (%)	14.9	28.6	18.2	6.1
**Peri-transplantation period**				
AKI (%)	58.1	85.7	63.6	48.5
RRT (%)	5.4	28.6	0	6.1
Surgical complications (%)	51.4	57.1	30.3	72.7
Transfusion requirements (%)	87.8	71.4	97	81.8
Vasoactive drugs (%)	54.1	85.7	98.5	54.5
Mechanical ventilation (%)	44.6	42.9	15.2	75.8
Infections (%)	62.6	57.1	36.4	90.9
Tacrolimus (%)	81.8	57.1	78.8	87.9
Cyclosporine (%)	23	42.9	18.2	24.2
Mycophenolate (%)	98.6	100	97	100
Everolimus (%)	8.2	14.3	9.1	6.1
Corticosteroids (%)	90.5	100	78.8	100
Changes in immunosuppression (%)	14.9	28.6	12.1	15.2
Graft rejection (%)	20.3	71.4	9.1	21.2
Calcineurin inhibitor overdose (%)	50	42.9	21.2	81.8
Nephrotoxic substances (%)	64.9	100	33.3	91
-NSAIDS (%)	4.16	14.2	9	0
-Antimicrobials (%)	31.27	42.9	63.66	16.7
-Intravenous contrast (%)	20.80	42.9	18.31	16.7
-Combinations of nephrotoxic (%)	43.77	0	9	66.6
Nephrologist Follow-up (%)	18.9	42.9	21.2	12.1
**Post-transplantation period**				
New hypertension (%)	23	28.6	15.1	30.3
PTDM (%)	23	28.6	15.2	30.3
New dyslipidemia (%)	17.5	57.1	12.1	18.2
Admissions (mean)	6.8	6.8	3.79	9.15
Infections	78.4	42.9	69.7	93.9
Calcineurin inhibitor overdose (%)	66.2	71.4	36.4	97
Changes in immunosuppression (%)	68.9	57.1	51.5	90.9
Graft rejection (%)	31.1	85.7	15.2	36.4
Nephrotoxic substances (%)	77	100	57.6	93.9
-NSAIDS (%)	1.75	0	5.2	0
-Antimicrobials (%)	17.53	0	42.84	6.49
-Intravenous contrast (%)	19.35	71.4	15.79	9.69
-Others (%)	1.75	0	5.2	0
-Combinations of nephrotoxic (%)	59.62	28.6	31.97	82.82
TMA (%)	9.5	14.3	0	18.2
**Final results**				
Time to Nephrologist follow-up (mean, months)	33.01	33.71	28.24	38.64
Worsened renal function (%)	45.9	28.57	44.11	51.5
Improved or stable renal function (%)	54.1	74.43	55.89	48.5
Increased baseline creatinine by 50% (%)	59.5	42.9	32.35	90.9
ESKD (%)	32.43	14.3	14.7	54.54
RRT (%)	10.8	14.3	6.1	15.2
Exitus (%)	28.4	0	32.35	30.3

Abbreviations: CKD: chronic kidney disease; AKI: acute kidney injury; RRT: renal replacement therapy; NSAIDs: non-steroidal anti-inflammatory drugs; PTDM: post-transplant diabetes mellitus; TMA: thrombotic microangiopathy; ESKD: end-stage kidney disease.

**Table 2 healthcare-11-01760-t002:** Statistically significant results of “increase baseline creatinine by 50%” variable.

Factors	n	%	*p*-Value	Hazard Ratio
Type of transplant				
Heart transplantation	3	42.9	*p* < 0.05	HR 0.075 [95% CI] 0.01 to 0.5
Liver transplantation	11	32.4	*p* < 0.05	HR 0.048 [95% CI] 0.012 to 0.192
Lung transplantation	30	90.9	*p* < 0.05	^¥^
**Pre-transplant period**				
CKD	3	17.6	*p* < 0.001	HR 0.084 [95% CI] 0.021 to 0.331
No follow-up by Nephrology	41	65.1	*p* < 0.027	HR 4.97 [95% CI] 1.196 to 20.651
**Peri-transplant period**				
Mechanical ventilation	27	81.8	*p* < 0.001	HR 6.353 [95% CI] 2.155 to 18.726
Calcineurin inhibitor overdose	28	75.7	*p* < 0.006	HR 4.083 [95% CI] 1.512 to 11.028
Nephrotoxic	33	68.8	*p* < 0.029	HR 3 [95% CI] 1.116 to 8.064
Antimicrobials	26	74.3	*p* < 0.012	HR 3.569 [95% CI] 1.324 to 9.62
Intravenous contrast	24	77.4	*p* < 0.009	HR 3.943 [95% CI] 1.403 to 11.082
No usage of everolimus	44	64.7	*p* < 0.001	HR 15722797292
No changes in immunosuppression	41	65.1	*p* < 0.027	HR 4.97 [95% CI] 1.196 to 20.651
No follow-up by Nephrology	39	65	*p* < 0.046	HR 3.34 [95% CI] 1.01 to 11.26
**Post-transplant period**				
Calcineurin inhibitor overdose	35	71.4	*p* < 0.004	HR 4.444 [95% CI] 1.594 to 12.390
Intravenous contrast	31	70.5	*p* < 0.022	HR 3.118 [95% CI] 1.182 to 8.226
Nº of hospital admissions	-	-	*p* < 0.006	HR 1.169 [95% CI] 1.046 to 1.306
Time until outpatient Nephrology consultation	-	-	*p* < 0.002	HR 1.032 [95% CI] 1.011 to 1.054

This table presents the analysis of various factors, including the type of transplant and different periods (pre-transplant, peri-transplant, and post-transplant), along with their corresponding percentages, *p*-values, and hazard ratios in the regression analysis. Significant associations with the dependent variable “increase baseline creatinine by 50%” are reported. A two-sided *p*-value < 0.05 was considered statistically significant. Abbreviations: CKD: chronic kidney disease. ^¥^ This parameter is set to zero because it is redundant.

**Table 3 healthcare-11-01760-t003:** Statistically significant results of ESKD. This table presents the analysis of various factors, including the type of transplant and different periods (pre-transplant, peri-transplant, and post-transplant), along with their corresponding percentages, *p*-values, and hazard ratios in the regression analysis. Significant associations with the dependent variable ESKD are reported. A two-sided *p*-value < 0.05 was considered statistically significant.

Factors	n	%	*p*-Value	Hazard Ratio
**Type of transplant**			*p* < 0.001	
Heart transplantation	1	14.3	*p* < 0.082	HR 0.13 [95% CI] 0.015 to 1.28
Liver transplantation	5	14.7	*p* < 0.001	HR 0.144 [95% CI] 0.045 to 0.463
Lung transplantation	18	54.5		This parameter is set to zero because it is redundant.
**Peri-transplant period**				
Mechanical ventilation	15	45.5	*p* <0.035	HR 2.963 [95% CI] 1.081 to 8.120
Calcineurin inhibitor overdose	17	45.9	*p* <0.015	HR 3.643 [95% CI] 1.27 to 10.372
Antimicrobials	16	45.7	*p* < 0.028	HR 3.158 [95% CI] 1.133 to 8.801
Intravenous contrast	16	51.6	*p* < 0.004	HR 4.667 [95% CI] 1.646 to 13.232
**Post-transplant period**				
Calcineurin inhibitor overdose	20	40.8	*p* < 0.037	HR 3.621 [95% CI] 1.078 to 12.161
Antimicrobials	18	42.9	*p* <0.032	HR 3.25 [95% CI] 1.106 to 9.548
Number of hospital admissions			*p* < 0.020	HR 1.103 [95% CI] 1.015 to 1.198

## Data Availability

The data presented in this study are available on request from the corresponding author. The data are not publicly available due to privacy and ethical reasons.

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
