# Peer review of "Role of the Nephrologist in Non-Kidney Solid Organ Transplant (NKSOT)"

_healthcare, 2023, doi:10.3390/healthcare11121760_

Round 1

Reviewer 1 Report

Viejo-Boyano present here a manuscript dubbed „ Role of the Nephrologist in Non-Kidney Solid Organ Transplant (NKSOT)“ that they submitted to MDPI-healthcare. I think the idea and the concept of the manuscript are really good and interesting to the readers of Healthcare. However, I have difficulties assessing the quality of the statistics. The authors do not provide any raw data or supplement that show how the calculations were performed. Further to me the material and methods section explaining the statistical tests is not exhaustive. To me, for instance, it is also not clear, where the authors used a T-Test on the data. I guess all the data ask for an ANOVA? But, as said, I am not an expert in statsistics, and I hope the second reviewer can give a more qualified comment on that.

Major finding:

Tables: Include a table legend above all tables. Write there what statistical methods were applied

 Figure 3: Describe the dataset more in the figure legend. n-numbers, etc. is there a statistical significant difference between the transplants? indicate in the figure.

Minor findings:

I would definitely recommend a proofreading by a native English speaker. There are several small typos throughout the text.

Line 22: hear = heart?

Line 24: nephrotoxicity?

Write all the m2 as m2

Line 132: a n is too much at the end oft he sentence

Line 135/136: 82%+23% is more than 100%? I guess there are some patients that can get both drugs, but as the sentence is written this is confusing

Line 136: nighty-nine is not a word. Nineteynine. However, I guess there is no need to write out the numbers greater than twelve.

Line 137: remove of by 82% of everolimus

Line 146: The mean number of hospital admissions was 6.8, with a maximum of 23 and a minimum of 1. : Here somehow the unit is missing.

Line 148. can you elaborate what kind of infections the patients had?

Line 163: „In 45 patients increased their baseline creatinine levels by 50%.“ Consider alternative wording

Line 228: remove „of“ from 21 of patients, after patients is a double space

Line 310: only the no uses of everolimus nor changes: revisit sentence

Author Response

Dear Reviewer,

Thank you for your valuable feedback and time on our work. We appreciate your positive remarks regarding the idea and concept of the manuscript, which we believe will be of great interest to readers.

Based on your comments, we acknowledge that additional clarification is required in the Materials and Methods section to enhance the understanding of our statistical analyses. We will make sure to provide more information that is comprehensive to address this concern.

See below our replies to your suggestions.

Q1. The authors do not provide any raw data or supplement that show how the calculations were performed. Further to me the material and methods section explaining the statistical tests is not exhaustive. To me, for instance, it is also not clear, where the authors used a T-Test on the data. I guess all the data ask for an ANOVA? But, as said, I am not an expert in statsistics, and I hope the second reviewer can give a more qualified comment on that.

A1. In our study, we analyzed various risk factors and their association with the categorical outcome variables. When the risk factors were categorical variables, we employed the Fisher's exact test when the cell frequencies in the contingency table were <5, and the chi-square test for larger sample sizes. For continuous risk factors, we first assessed normality using the Kolmogorov-Smirnov test. If the continuous variables followed a normal distribution, we utilized the Student's t-test. Alternatively, if normality assumptions were violated, we employed the Mann-Whitney U test. Furthermore, we used the Pearson correlation coefficient for assessing associations between normally distributed continuous variables and the Spearman correlation coefficient for continuous variables that did not follow a normal distribution. We also utilized the Pearson correlation coefficient as a measure of association (also known as the coefficient of contingency) for categorical data.

To analyze the relationship between nephrotoxics substances and the categorical outcome variable, we performed multinomial regression analyses instead of ANOVA, as the final outcome variable was categorical. Lastly, we performed survival analysis using the Kaplan-Meier method to assess the survival rates for each type of transplant in relation to the outcome variables.

We believe that incorporating these additional details in the revised Materials and Methods section will provide a more thorough and transparent explanation of our statistical approaches.

Q2. Tables: Include a table legend above all tables. Write there what statistical methods were applied. Figure 3: Describe the dataset more in the figure legend. n-numbers, etc. is there a statistical significant difference between the transplants? indicate in the figure.

A2. Following your and other reviewer valuable suggestion we made other changes as follow: Tables: We have included a table legend above each table, clearly stating the statistical methods applied for analyzing the data in each table. This provides readers with a better understanding of the analytical approaches used in our study. Figure 3: We have updated the figure legend to provide a more comprehensive description of the dataset.

Q3. I would definitely recommend a proofreading by a native English speaker. There are several small typos throughout the text.

A3. We addressed this minor finding regarding proofreading. We have thoroughly proofread the manuscript and made necessary corrections to ensure proper grammar, syntax, and clarity. We appreciate the suggestion and apologize for any previous typos that may have affected the readability of the manuscript. Finally, we believe these revisions have significantly improved the manuscript and addressed the reviewer's comments.

Reviewer 2 Report

1. section  "3. Results, 3.1. Baseline characteristics":

- there are many data, presented in text and in table 1 as well - this is duplication of data presentation and shiould be omitted.

2. "3.2. Increased baseline creatinine by 50%"

Table 2. Statistically significant results of Increase baseline creatinine by 50% variable.  

- the word "creatinine" is missing in the Table 2 Heading.

Again, there are many data, presented in text and in table 2 as well which is duplication of data presentation 

3. Authors should write the study limitations at the end, especially a relatively small sample size and, therefore,  comment on  power of statistical methods used.

Author Response

Dear reviewer, we really appreciate and thank for your valuable effort and time spent on revising our work. Below we provide a reply to each of your comments, which have allowed for a clearer and a more synthetic work.

Q1. 1. Section 3. Results, 3.1. Baseline characteristics": there are many data, presented in text and in table 1 as well - this is duplication of data presentation and should be omitted.

A1. We appreciate your concern about potential duplication of data. We understand the need for clarity and comprehensive reporting in our manuscript. While some data may overlap between the text and the table 1, we believe it is essential to provide both the overall results and the specific results for each transplant type. The text summarizes the global findings, while Table 1 allows for a detailed comparison across the different transplant groups, enabling readers to discern any distinct patterns or differences.

To address your concern and improve the manuscript's readability, we will ensure that the data presented in the text and Table 1 are appropriately cross-referenced, allowing readers to seamlessly navigate between the two sources of information. This approach will eliminate redundancy while providing a comprehensive understanding of the results. Nevertheless, we have summarized the text and clarified Table 1 by adding abbreviations and slightly changing the structure of the table.  

Q2. 3.2. Increased baseline creatinine by 50%"

A2. Thank you, it has already been corrected.

Q3. Table 2. Statistically significant results of increase baseline creatinine by 50% variable: the word "creatinine" is missing in the Table 2 Heading; again, there are many data, presented in text and in table 2 as well which is duplication of data presentation 

A3. Thank you for pointing out the missing word "creatinine" in the Table 2 heading. We have rectified this oversight and ensured that the corrected version accurately represents the variable under analysis.

Regarding the duplication of data presentation, we understand the concern and have made appropriate revisions to avoid redundancy. In the text, we have provided a concise summary of the key statistical points, highlighting the most important findings. On the other hand, in Table 2, we have emphasized the statistical significance of each variable, allowing readers to easily identify and interpret the significant results.

By presenting the information in this manner, we aim to strike a balance between providing an overview in the text and presenting the detailed statistical significance in the table. This approach ensures that readers can efficiently grasp the main findings while also having access to the specific statistical details in the table.

Q4. Authors should write the study limitations at the end, especially a relatively small sample size and, therefore, comment on power of statistical methods used.

A4. We thank your valuable feedback. As per your suggestion, we have included a dedicated section at the end of the discussion that addresses the limitations of our study. We have discussed the small sample size, which may limit the generalizability of our findings, and the single-center design, which introduces potential biases and may restrict the external validity of the results. By acknowledging these limitations, we aim to provide a more comprehensive interpretation of the aim of our study and encourage further research in the field.

Reviewer 3 Report

This paper aims to provide a single-center retrospective observational study on a cohort of Chronic kidney disease (CKD) patients under follow-up in the department of nephrology at La Fe University and Politècnic Hospital in Valencia (Spain) from January 2010 to December 2020. Four dependent variables were chosen, and statistical analysis was performed between a set of risk factors and these four dependent variables: ESKD, increased serum creatinine ≥50%, renal replacement therapy, and death to assess which risk factor is significant correlated to the dependent variable. This study is generally sound with the following minor concerns:

  1. Test Design:

It seems that author is trying to extend a prior research performed in 2003 using 1990 till 2000 data. But, based on the description provided within this paper, the 2003 study only analyzed one of the four dependent variables in the current study (i.e. developed End Stage Kidney Disease after a non-kidney solid organ transplant). However, this study has expended the analysis to additional three dependent variables. Could author please add more descriptive language to provide more background on the dependent variable selections.

  1. Statistical Analysis:

Multinomial regression was leveraged as part of this study. However, it seems that author did not perform multicollinearity analysis with the independent variables, especially for section 3.2, where more than 10 variables were statistically significant. Multicollinearity analysis should be performed to support the final conclusion.

  1. The authors should have a section discussed about the limitation of the analysis and study, especially when the sample size is relatively small.

Author Response

Dear reviewer,

First we want to extent our regards and gratitude for yours valuable suggestion and time. Please see below our replies to your suggestions regarding our work.

Q1. It seems that author is trying to extend a prior research performed in 2003 using 1990 till 2000 data. But, based on the description provided within this paper, the 2003 study only analyzed one of the four dependent variables in the current study (i.e. developed End Stage Kidney Disease after a non-kidney solid organ transplant). However, this study has expended the analysis to additional three dependent variables. Could author please add more descriptive language to provide more background on the dependent variable selections.

A1. We are not attempting to extend a prior research conducted in 2003, but rather highlighting that the aforementioned study represents the largest analysis of patients with non-kidney solid organ transplant (NKSOT) and kidney disease to date. However, in our study, we aimed to investigate not only the development of advanced CKD but also other clinically relevant variables for nephrologists, such as doubling of serum creatinine, the need for renal replacement therapy, and mortality. By including these additional dependent variables, we aimed to provide a comprehensive understanding of the impact of NKSOT on kidney-related outcomes.

Q2. Multinomial regression was leveraged as part of this study. However, it seems that author did not perform multicollinearity analysis with the independent variables, especially for section 3.2, where more than 10 variables were statistically significant. Multicollinearity analysis should be performed to support the final conclusion.

A2. We appreciate the reviewer's concern regarding the absence of multicollinearity analysis. However, in our study, the analysis of multicollinearity was not performed due to the unique nature of the different periods (pre-transplant, peri-transplant, and post-transplant) and the variables examined. As previously mentioned, we aimed to analyze the associations and effects of the risk factors separately within each period. Of note, we performed a specific multivariable regression analysis focusing on nephrotoxic agents, including antimicrobials, intravenous contrast, and NSAIDs. This analysis aimed to investigate the individual contributions of these agents in terms of their association with the outcomes of interest. We found that among these globally considered nephrotoxic agents, specifically in the section mentioned, the use of intravenous contrast emerged as a significant risk factor, indicating a potential increased risk for the development of adverse renal outcomes.

Q3. The authors should have a section discussed about the limitation of the analysis and study, especially when the sample size is relatively small.

A3. We appreciate the reviewer's valuable insights regarding the limitations of our study. We included a paragraph in the revised manuscript addressing these limitations. Specifically, we acknowledge that the small sample size of 74 patients may restrict the generalizability of our findings. We emphasize the need for larger sample sizes to enhance statistical power and improve the representativeness of the results. Additionally, we acknowledge the single-center design as a potential source of bias and limitations in terms of external validity. We recognize the importance of replicating the study in multiple centers to strengthen the overall findings and enhance their generalizability to a broader population. Finally, thank you for highlighting these aspects, and we have taken appropriate steps to address these limitations in the revised version of the manuscript.

Reviewer 4 Report

Thank you very much for sharing this article. I was asked to strictly review this article as a statistician, as I am unfamiliar with this research topic. That being said, I have several minor comments.

1) Please provide more details on the sample size for each analysis and specify the criteria used to determine whether to use the Chi-square test or the Fisher's test for qualitative variables, as well as the Student's T test or the Mann-Whitney U test for quantitative variables. This information will help readers better understand the rationale behind the chosen statistical tests.

2) It would be helpful to briefly explain the Kolmogorov-Smirnov normality test and its results for the quantitative variables. Additionally, clarify whether the test was applied to the entire dataset or specific subgroups.

3) Clarify the reason for choosing either Pearson or Spearman correlation for the evaluation of statistically significant variables. Provide the normality parameters used to guide the selection of the correlation method.

4) When describing the Kaplan-Meier statistic for survival analysis, consider providing additional details such as the outcome variable(s) examined, the censoring criteria, and the length of the follow-up period. In addition, HRs have been reported, indicating that the authors have used some parametric models. Please discuss the choice of your models.

5) It would be valuable to include information on any adjustments made for multiple testing to address the inflated Type I error rate issue.

Author Response

Dear reviewer, many thanks for your help and suggestions to improve our work. Following your and other reviewers' suggestions, we made several changes to our work. Nonetheless, we will appreciate it if you consider in a positive way our replies in this issue.

Our study analyzed various risk factors and their association with the categorical outcome variables. When the risk factors were categorical variables, we employed Fisher's exact test when the cell frequencies in the contingency table were <5, and the chi-square test for larger sample sizes. For continuous risk factors, we first assessed normality using the Kolmogorov-Smirnov test. If the continuous variables followed a normal distribution, we utilized the Student's t-test. Alternatively, if normality assumptions were violated, we employed the Mann-Whitney U test.

Furthermore, we used the Pearson correlation coefficient for assessing associations between normally distributed continuous variables and the Spearman correlation coefficient for continuous variables that did not follow a normal distribution. We also utilized the Pearson correlation coefficient as a measure of association (also known as the coefficient of contingency) for categorical data.

In the Kaplan-Meier analysis, the outcome variables examined were the development of advanced chronic kidney disease and the need for renal replacement therapy, with the independent variable being the type of transplant. Censoring criteria were apply to individuals who did not develop advanced CKD or RRT and completed the follow-up period. The mean follow-up duration was 59.61 months, with a maximum follow-up time of 142 months. Regarding the use of HR, they were employ in the regression analysis to assess the risk of developing the dependent variables across different types of transplants.

Round 2

Reviewer 3 Report

issues from the first round of review were addressed